# Profile of congenital heart disease in infants born following exposure to preeclampsia

Christopher S. Yilgwan[1]*, Victor C. Pam[2], Olukemi O. Ige[1], Williams N. Golit[3], Stephen Anzaku[4], Godwin E. Imade[2], Gavou Yilgwan[5], Josiah T. Mutihir[2], Atiene S. Sagay[2], Augustine Odili[6], Ayuba I. Zoakah[6], Fidelia Bode-Thomas[1], Melissa A. Simon[7,8]

**1** Department of Paediatrics, University of Jos, Jos, Nigeria, **2** Department of Obstetrics and Gynaecology, University of Jos, Jos, Nigeria, **3** Department of Obstetrics and Gynaecology, Plateau Specialist Hospital, Jos, Nigeria, **4** Department of Obstetrics and Gynaecology, Bingham University, Karu LGA, Nigeria, **5** Department of Human Physiology, University of Jos, Jos, Nigeria, **6** Department of Internal Medicine, University of Abuja, Abuja, Nigeria, **7** Department of Community Medicine, University of Jos, Jos, Nigeria, **8** Department of Obstetrics and Gynecology, Northwestern University, Chicago, IL, United States of America

* yilgwan@hotmail.com, yilgwanc@unijos.edu.ng

## Abstract

### Background

Events in pregnancy play an important role in predisposing the newborn to the risk of developing CHD. This study evaluated the association between maternal preeclampsia and her offspring risk of CHD.

### Methods

This is a cohort study of 90 sex-matched neonates (45 each born to women with preeclampsia and normal pregnancy) in Jos, Nigeria. Anthropometry was taken shortly after delivery using standard protocols. Echocardiography was performed within 24 hours of life and repeated 7 and 28 days later. SPSS version 25 was used in all analyses. Statistical significance was set at p<0.05.

### Results

Congenital heart disease (CHD) was observed in 27 (30.0%) of newborns of women with preeclampsia compared with 11 (12.1%) of newborns without preeclampsia (p<0.001) at the end of 7 days and in 19 (21.1%) of newborns of women with preeclampsia and 3 (3.3%) of newborns of women without preeclampsia by the end of the 4th week of life (p<0.001). Overall, ASD (4 newborns), PDA (21 newborns), patent foramen ovale (14 newborns) and VSD (2 newborns) were the prevalent lesions found among all the newborns studied in the first week of life. Isolated atrial and ventricular septal defects were seen in 4 (4.4%) of the newborns of women with preeclampsia. Being the infant of a woman with preeclampsia was associated with about 8-fold increased risk of having CHD (OR = 7.9, 95% CI = 2.5–24.9, p<0.001).

**Data Availability Statement:** The data will be held in a public repository. the URL link to the data is: https://doi.org/10.6084/m9.figshare.10086140

**Funding:** CSY, VCP and OOI received mentorship training grant from STAMINA D43TW010130 grant. The funders had no role in study design, data collection and analysis, decision to publish, or preparation of the manuscript.

**Competing interests:** The authors have declared that no competing interests exist

## Conclusion

CHD may be more common in newborns of women with preeclampsia underscoring the need for fetal and newborn screening for CHD in women with preeclampsia so as to improve their infant's well being.

## Introduction

Preeclampsia/eclampsia (PE/E) has remained a significant public health problem in Nigeria and the developing world. It is said to complicate an estimated 2–10% of all pregnancies resulting in five-fold increase in perinatal morbidity and mortality.[1,2] The pathophysiology of PE/E entails generalized endothelial dysfunction initiated by abnormal placentation. This endothelial dysfunction is associated with different degrees of fetal injury even though perinatal outcome is also influenced by gestational age and severity of the hypertension.[3,4] Preeclampsia is thought to result from disturbed placental function in early pregnancy possibly due to failed interaction between two genetically different organisms. [4,5] As early as 12 weeks gestation, placental flow defects have been shown to occur in women who subsequently develop pre-eclampsia. This placental dysfunction causes flow defect and hypoxia episodes with resultant generation of reactive oxygen species, placental oxidative stress and increased serum fms-like tyrosine kinase 1(sFlt-1)production similar to what is seen in human fetuses with congenital heart diseases. [4,5] The hypoxia induced overproduction of sFlt-1 could activate a cycle wherein high sFlt-1 levels inhibit angiogenesis and exacerbate the placental hypoxia, which subsequently causes an increase in placental sFlt-1 production.[4,5] Considering this similarity in pathogenesis, it is possible that angiogenic imbalance may play an important role in the pathogenesis of CHD especially in infants of women with preeclampsia.[3,5]

Congenital heart diseases (CHD) are the most common birth defects globally.[6,7] They are also important cause of morbidity and mortality globally.[8,9] Current estimates by the Centers for Disease Control and Prevention (CDC) in the US report CHD as responsible for about 4.2% of all neonatal deaths with most occurring in the first 28 days of life. Apart from genetic (e.g. trisomies), infectious (e.g. TORCHES), drugs, alcohol and environmental factors, febrile illness and inflammation have also been shown to predispose infants to CHD.[10]

Preeclampsia is associated with poor infant outcomes especially low birth weight and prematurity.[11] Being an inflammatory disorder, preeclampsia itself may predispose infants to congenital anomalies including CHD.[11,12] It is thus relevant to investigate the possible association between CHD and preeclampsia. We therefore sought to characterize the spectrum and burden of CHD in infants born following preeclampsia and also to determine whether preeclampsia is associated with CHD.

## Materials and methods

### Study setting

We conducted this study in the 4 tertiary health facilities in Jos namely; Jos University Teaching Hospital (JUTH), Plateau Specialist Hospital (PSSH), Bingham University Teaching Hospital (BhUTH) and Our Lady of Apostle (OLA) Hospital. Each of these hospitals has specialist obstetric care services. Altogether, they have an annual delivery rate of about 14,000 babies with preeclampsia/eclampsia accounting for about 5% of the deliveries annually.

## Study population

This study was carried out between April 2017 and May 2018 as part of the infant outcomes study on women with preeclampsia in Jos. All the women were recruited antenatally and followed up to delivery. At delivery, all the infants had anthropometry and echocardiography done. We identified 45 newborns from women with preeclampsia at birth in the delivery room and matched them for sex with 45 newborns of women with normal pregnancy. We used OpenEpi version 3.03a to determine a minimum sample size of 80 (40 neonates in each arm) based on the estimated effect size of 20%, a power of 80% and an α level of 0.05.[13] We excluded newborns of women with other chronic disorders like diabetes, HIV and Sickle cell anemia in the control and exposure groups in order to avoid confounding. Ethical approval was obtained from each of the participating hospitals before commencement of the study. Written informed consent was obtained from the mothers before recruiting the newborns.

Preeclampsia was defined as systolic blood pressure ≥140 mmHg or diastolic pressure ≥90 mmHg (or increases of 30 mmHg systolic or 15 mmHg diastolic from the baseline) on at least two occasions, six or more hours apart and associated proteinuria that develops from the 20th gestational week in a previously normotensive woman.

## Study design

This study was a cohort design that compared the spectrum of CHD in newborns of women with preeclampsia versus those with normal pregnancy in the four tertiary care centers in Jos Nigeria.

## Study procedure

Each newborn had a transthoracic echocardiography done at least 4 hours post-delivery in the nursery to assess for the presence of CHD. This was repeated for all the infants at 7 days and 28 days of life in order to exclude physiologic patent ductus arteriosus and foramen ovale. All measurements were performed according to American Society of Echocardiography guidelines by an experienced Paediatric Cardiologist.[14] Echocardiography was done following a standard examination protocol using a Vivid e ultrasound machine (General Electric, USA) equipped with a P6 Phased Array ultrasound transducer.[14] Anthropometric measurements were done by weighing each naked newborn to the nearest 50g using a way master bassinet weighing scale operated by a trained midwife. [15,16]

## Ethical considerations

The institutional review board of the Jos University Teaching Hospital reviewed and approved the study (JUTH/DCS/ADM/127/XIX/6632).

Each infant was identified with a code that does not contain their name. Only the principal investigator has access to the database linking the name of the individual with the code. All material data including the study forms and specimen bottles were labeled accordingly with the printed unique identification numbers. Different biological fluids (i.e. serum and plasma specimen) were marked with different prefixes for ease of identification.

## Statistical analysis

Statistical analysis was done using SPSS version 25.[17] Mean differences in maternal age, booking weight and parity as well as infant weight and gestational age were compared between babies born following preeclamptic pregnancy and those born following normal pregnancy using a t-test. Difference in proportion of the newborns by sex as well as maternal education,

**Table 1. Baseline demographics and clinical characteristics of the study population (N = 90).**

| Variable | Preeclampsia (n = 45) | Normal Pregnancy (n = 45) | t | p |
|---|---|---|---|---|
| **Mean Age (years)** | 31.1±6.3 | 29.3±5.6 | 1.40 | 0.17 |
| **Mean Booking weight (kg)** | 73.9±19.0 | 64.5±14.0 | 2.50 | *0.02* |
| **Mean Maternal BMI(Kg/m$^2$)** | 46.3±11.6 | 40.1±8.3 | 2.69 | *0.009* |
| **Mean Parity** | 2.9±2.0 | 2.9±2.2 | 0.02 | 0.99 |
| **Maternal Education** | | | | |
| *At least Primary School* | 11(24.4%) | 7(15.6%) | 1.24 | 0.57σ |
| *Secondary School* | 13(28.8%) | 13(28.8%) | | |
| *Beyond Secondary School* | 21(46.6%) | 25(55.6%) | | |
| **Fever in Pregnancy** | 11(24.4%) | 8(17.8%) | 23.64 | *<0.001σ* |
| **Infant Sex** | | | | |
| *Male* | 25(55.6%) | 25(55.6%) | 3.34 | 0.18σ |
| *Female* | 20(44.4%) | 20(44.4%) | | |
| **Mean Birth Weight (grams)** | 2529.5±817.5 | 3079.2±527.4 | 3.79 | *<0.001* |
| **Mean Gestational Age (weeks)** | 36.8±3.2 | 38.7±2.0 | 3.35 | *0.001* |

σ = Chi square statistics, significant values are in italics

Because fewer than 5 women per group reported alcohol and cigarette intake, we excluded those variables from the analysis.

fever in pregnancy, alcohol use and contraceptive use was done using 2 by 2 table cross tabulations. Spectrum of CHD was depicted using bar charts and a frequency table. Univariate analysis was done to evaluate the relationship of each of the maternal and infant variables with CHD. Those that were found to be significant were then included in Multivariable logistic regression analysis to assess the relationship between CHD and these maternal and infant characteristics. The criterion for significance for all analyses was set at a P-value of < 0.05.

## Results

### Descriptive characteristics

Table 1 shows maternal and infant descriptive characteristics. There were 25 (55.6%) male newborns and 20 (44.4%) female newborns delivered to both women with preeclampsia and those with normal pregnancy. The newborns of women with preeclampsia were significantly lighter compared with the newborns of women with normal pregnancy (mean birth weight 2529.5±817.5 vs 3079.2±527.4 grams; p<0.001) and of younger gestational age (36.8 ± 3.2 vs 38.7±2.0 weeks; p = 0.001). Women with preeclampsia were significantly heavier compared with women without preeclampsia (mean weight at booking of 73.9±19.0 kg vs 64.5±14.0 kg; p = 0.02, mean BMI at booking 46.3±11.6 vs 40.1±8.3, p = 0.009)), but not significantly older (31.1± 6.3 vs 29.3±5.6 years; p = 0.17). Mean maternal parity was not significantly different between the two groups (preeclampsia = 2.9±2.0 vs normal pregnancy = 2.9±2.2, p = 0.99), whereas fever in pregnancy was significantly more common in the preeclampsia group compared to women with normal pregnancy (11[12.2%] vs 8[8.9%]; p<0.001).

### Prevalence and spectrum of congenital heart disease (CHD)

As shown in Fig 1A & 1B and Table 2, by day 7, CHD was observed in 27 (60.0%) of newborns of women with preeclampsia compared with 11(22.2%) of newborns of women with normal pregnancy (p<0.001). By day 28 of life, 19 (42.2%) of newborns of women with preeclampsia had CHD, while 3 (6.6%) of newborns of women with normal pregnancy had CHD.

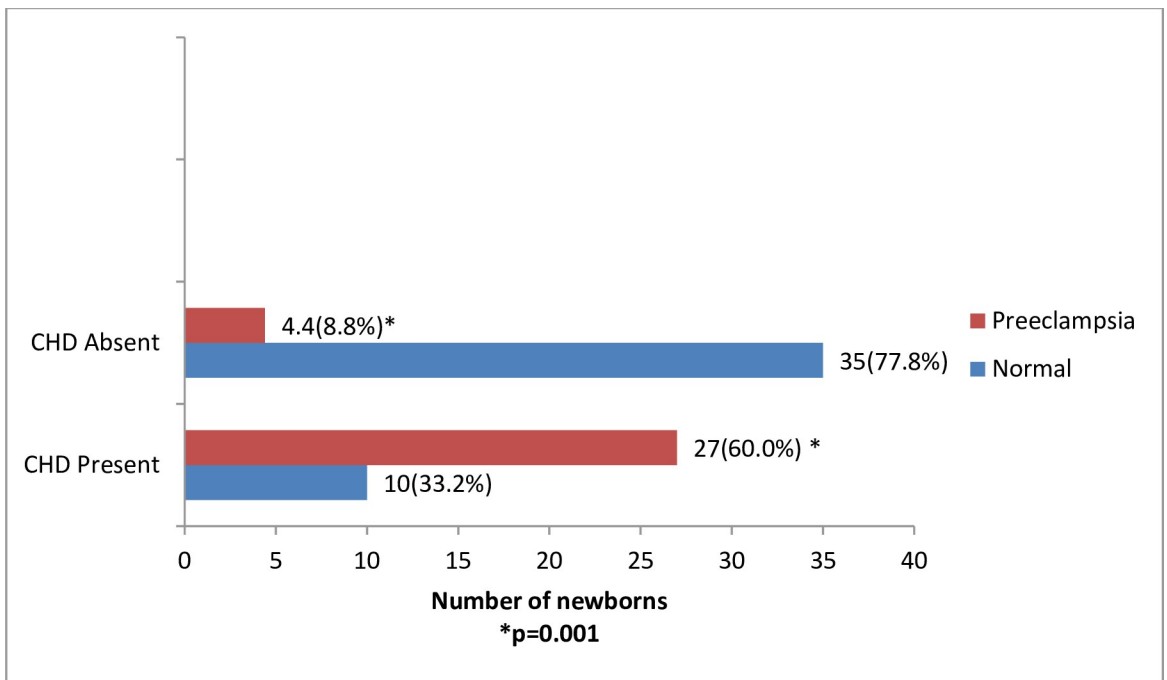

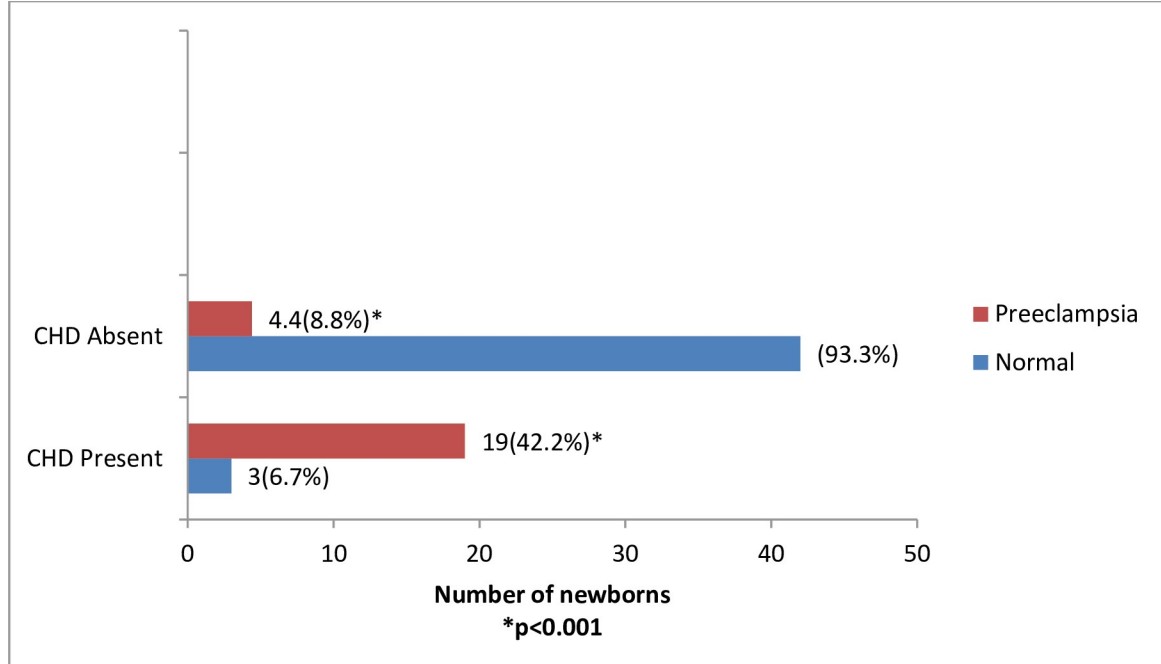

**Fig 1. (A)** Distribution of Congenital Heart Disease (CHD) in newborns by preeclampsia exposure status at the end of day 7. **(B)** Distribution of Congenital Heart Disease (CHD) in newborns by preeclampsia exposure status at the end of day 28.

Atrial septal defect (ASD), patent ductus arteriosus (PDA), patent foramen ovale (PFO) and ventricular septal defects (VSD) were the prevalent CHD types observed which persisted beyond the first week of life. Isolated ASD and VSD were observed only in newborns born to women with preeclampsia (Table 2). The PDA in newborn of normal pregnancy and 8 (8.9%) of those newborns of preeclamptic women had closed by the 28th day of life (Table 2).

**Table 2. Frequency of occurrence of type of congenital heart disease (CHD) in the study population (N = 90) by exposure status.**

| | Day 7 | | Day 28 | |
|---|---|---|---|---|
| Variable | Preeclampsia n = 45 | Normal pregnancy n = 45 | Preeclampsia n = 45 | Normal pregnancy n = 45 |
| ASD | 4(8.8%) | 0(0.0%) | 4(8.8%) | 0(0.0%) |
| Isolated | 2(4.4%) | 0(0.0%) | 2(4.4%) | 0(0.0%) |
| With PDA | 2(4.4%) | 0(0.0%) | 2(4.4%) | 0(0.0%) |
| PDA | 13(28.8%) | 8(17.8%) | 4(8.8%) | 1(2.2%) |
| Isolated | 8(17.8%) | 7(7.8%) | 2(2.2%) | 1(1.1%) |
| With PFO | 3(6.6%) | 1(2.2%) | 0(0.0%) | 0(0.0%) |
| With ASD | 2(4.4%) | 0(0.0%) | 2(2.2%) | 0(0.0%) |
| PFO | 11(24.4%) | 3(6.6%) | 11(24.4%) | 2(4.4%) |
| VSD | 2(4.4%) | 0(0.0%) | 2(4.4%) | 0(0) |
| *Total | 27(60.0%) | 10(22.2%) | 19(42.2%) | 3(6.6%) |

*due to 2 patients having a combination of lesions, the frequency does not sum.

## Gestational age distribution by CHD status

In Fig 2, CHD is seen mainly among newborns with gestational age between 35–40 weeks. The gestational age distribution of those newborns with CHD is similar to that of newborns without CHD.

## Factors associated with congenital heart disease (CHD)

Table 3 shows logistic regression coefficients of variables associated with CHD. Being the infant of a woman with preeclampsia was associated with an 8-fold increased likelihood of having CHD (OR = 7.9, 95%CI = 2.5–24.9, p<0.001). Being a female infant or being preterm was

**Fig 2. Distribution of gestational age by CHD status.**

**Table 3. Logistic regression estimates of the relationship between some study variables and congenital heart disease (CHD).**

| Variables | B | SE | Wald | p value | Exp(B) | 95% CI | |
|---|---|---|---|---|---|---|---|
| | | | | | | Lower | Upper |
| Birth weight | .00 | .001 | .424 | .52 | 1.0 | 1.0 | 1.0 |
| Miscarriages | .28 | .299 | .892 | .33 | 1.3 | 0.7 | 2.4 |
| *Preeclampsia* | *2.06* | *.586* | *12.437* | *< .001* | *7.9* | *2.5* | *24.9* |
| Maternal age(yr) | -.03 | .045 | .556 | .456 | .97 | .89 | 1.1 |
| Gestational age | .10 | .144 | .563 | .45 | 1.1 | .84 | 1.5 |
| Female infant | .56 | .523 | 1.145 | .29 | 1.8 | .63 | 4.9 |
| Prematurity | .36 | .953 | .142 | .70 | 1.4 | .22 | 9.3 |

CI = Confidence interval, SE = Standard error

associated with an increased likelihood of having CHD of 1.8 times (OR = 1.8, 95%CI = 0.63–4.9, p = 0.29) and 1.4 times (95% CI = 0.22–9.3, p = 0.70) respectively though not statistically significant.

## Discussion

We found a significantly higher proportion of newborns born to women with preeclampsia having CHD compared to those born to women with normal pregnancy. Shunt lesions such as isolated ASD and VSD were predominantly seen in newborns of women with preeclampsia. Patent ductus arteriosus and PFO were isolated in both groups of newborns. This is similar to what Auger and colleagues[18] reported in an analysis of hospital discharge records in Quebec Canada, where a high prevalence of CHD was observed in women with preeclampsia compared with normal pregnancy. Similar to our findings, Auger reported a high proportion of shunt lesions like ventricular septal defects as the most common CHD isolated. [18]

Though prematurity is associated with a higher prevalence of CHD globally, our study did not find a significant association between prematurity and the odds of having CHD.[19] This may be due to the fact that PDA, a common CHD in premature infants was only considered in the diagnosis after the 4[th] week of life so as to exclude those PDA that may be as a result of delayed closure of the ductus arteriosus in premature infants. [19,20] In addition, we did not find any significant association between gender and the risk of having CHD similar to global reports.[20,21]

We found a nearly 8-fold increased risk of CHD in the newborns of women with preeclampsia. Current reports suggest an association between preeclampsia and the heightened risk of CHD in fetuses. In a study of a cohort of 700,000 Danish women, Boyd et al.[21] reported that early preeclampsia was associated with a 6-fold increased CHD risk in offspring. In addition, pregnant women whose fetuses have been shown to have CHD *in utero* have been shown to have a nearly 7-fold increased risk of developing preeclampsia, regardless of the nature of the CHD.[21] The increased risk for developing CHD in fetuses of women with preeclampsia has been attributed to the likelihood of shared angiogenic imbalance between the mother and fetus. [21–23] Since studies have shown a causal relationship between angiogenic imbalance and occurrence of developmental defects of the human fetal heart, it is thus not surprising finding an increased risk of CHD in the newborns of the women with preeclampsia in our study.[22,23]

### Clinical significance of our study results

With the advent of echocardiography and the development of fetal and newborn screening protocols, early and timely diagnosis of CHD is possible and has improved not just the

management of these newborns but also their survival.[24,25] The association between pre-eclampsia and CHD may suggest that clinicians managing women with preeclampsia in addition to other fetal surveillance screening, consider including the screening for congenital heart diseases in order to improve the early diagnosis and management of these newborns who may have CHD. This will help improve the survival of the infants of women with preeclampsia. [18,24,25]

## Strengths of our study results

Strengths of our study results include the cohort design of the study, the collaborative efforts between obstetricians and paediatricians, and the use of standard definitions and protocols for the classification and diagnosis of CHD and preeclampsia. In addition, the inclusion of a 28 day follow up on the newborns allowed us to exclude those lesions such as patent ductus arteriosus and patent foramen ovale, which are transiently present at birth as a result of physiologic changes in newborns.

## Limitations of our findings

The small sample size, while appropriately powered for a pilot study and producing statistically significant results, is a limitation of the present study. In addition, we did not include abortuses and stillborns delivered in the study because of the inability to perform fetal and neonatal autopsy. The exclusion of CHD from these sources (abortuses, still births) may have affected the potential relationship between preeclampsia and CHD reported here. A large population based study incorporating pathological examination of stillborns and abortuses would thus be needed to validate our findings.

## Conclusions

Congenital heart diseases may be more common in newborns of women with preeclampsia. This underscores the need for fetal and newborn screening for congenital heart defects as part of the fetal and newborn surveillance studies in women with preeclampsia so as to improve their infant's well being.

## Supporting information

**S1 Checklist. *PLOS ONE* clinical studies checklist.**
(DOC)

**S1 Dataset.**
(XLS)

## Acknowledgments

We acknowledge the support and contributions of the nurses in Jos University Teaching Hospital, Bingham University Teaching Hospital, Our Lady of Apostles Hospital and Plateau State Specialist Hospital, Jos.

## Author Contributions

**Conceptualization:** Christopher S. Yilgwan, Gavou Yilgwan, Atiene S. Sagay, Ayuba I. Zoakah, Fidelia Bode-Thomas.

**Data curation:** Christopher S. Yilgwan, Victor C. Pam, Olukemi O. Ige, Williams N. Golit, Stephen Anzaku, Godwin E. Imade.

**Formal analysis:** Christopher S. Yilgwan, Godwin E. Imade, Gavou Yilgwan, Ayuba I. Zoakah.

**Funding acquisition:** Christopher S. Yilgwan.

**Investigation:** Christopher S. Yilgwan, Victor C. Pam, Olukemi O. Ige, Williams N. Golit, Godwin E. Imade, Gavou Yilgwan.

**Methodology:** Christopher S. Yilgwan, Victor C. Pam, Stephen Anzaku, Godwin E. Imade, Gavou Yilgwan, Josiah T. Mutihir, Atiene S. Sagay, Augustine Odili, Ayuba I. Zoakah, Fidelia Bode-Thomas, Melissa A. Simon.

**Project administration:** Christopher S. Yilgwan, Victor C. Pam.

**Resources:** Christopher S. Yilgwan, Williams N. Golit, Stephen Anzaku, Josiah T. Mutihir, Augustine Odili.

**Software:** Christopher S. Yilgwan.

**Supervision:** Godwin E. Imade, Josiah T. Mutihir, Atiene S. Sagay, Ayuba I. Zoakah, Fidelia Bode-Thomas, Melissa A. Simon.

**Validation:** Christopher S. Yilgwan, Godwin E. Imade, Josiah T. Mutihir.

**Visualization:** Josiah T. Mutihir.

**Writing – original draft:** Christopher S. Yilgwan, Stephen Anzaku, Gavou Yilgwan, Josiah T. Mutihir, Atiene S. Sagay, Augustine Odili, Ayuba I. Zoakah, Fidelia Bode-Thomas, Melissa A. Simon.

**Writing – review & editing:** Christopher S. Yilgwan, Victor C. Pam, Olukemi O. Ige, Williams N. Golit, Stephen Anzaku, Godwin E. Imade, Gavou Yilgwan, Josiah T. Mutihir, Atiene S. Sagay, Augustine Odili, Ayuba I. Zoakah, Fidelia Bode-Thomas, Melissa A. Simon.

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
