## [Decision Letter · Decision Letter 0]

14 Jan 2020

PONE-D-19-30623

PROFILE OF CONGENITAL HEART DISEASE IN INFANTS BORN FOLLOWING EXPOSURE TO PREECLAMPSIA

PLOS ONE

Dear Dr. Yilgwan,

Thank you for submitting your manuscript to PLOS ONE. After careful consideration, we feel that it has merit but does not fully meet PLOS ONE’s publication criteria as it currently stands. Therefore, we invite you to submit a revised version of the manuscript that addresses the points raised during the review process.

We would appreciate receiving your revised manuscript by 03/01/2020. To enhance the reproducibility of your results, we recommend that if applicable you deposit your laboratory protocols in protocols.io, where a protocol can be assigned its own identifier (DOI) such that it can be cited independently in the future. For instructions see: http://journals.plos.org/plosone/s/submission-guidelines#loc-laboratory-protocols

We look forward to receiving your revised manuscript.

Kind regards,

Linglin Xie

Academic Editor

PLOS ONE

Journal Requirements:

Dr Christopher Yilgwan was supported under the STAMINA Mentorship grant by the Fogarty

International Center (FIC), the NIH Common Fund, Office of Strategic Coordination, Office of

the Director (OD/OSC/CF/NIH) Office of AIDS Research, Office of the Director (OAR/OD),

National Institute of Neurological Disorders and Stroke, Office of the Director (NINDS/NIH),

and the National Institute of Nursing Research(NINR/NIH) of the National Institutes of Health

under award numbers D43TW010130.

CSY, VCP and OOI received mentorship training grant from STAMINA D43TW010130 grant. The funders had no role in study design, data collection and analysis, decision to publish, or preparation of the manuscript.

5. Please remove your figures from within your manuscript file, leaving only the individual TIFF/EPS image files, uploaded separately.  These will be automatically included in the reviewers’ PDF.

Reviewers' comments:

Reviewer's Responses to Questions

**Comments to the Author**

1. Is the manuscript technically sound, and do the data support the conclusions?

Reviewer #1: Yes

Reviewer #2: Yes

2. Has the statistical analysis been performed appropriately and rigorously? 

Reviewer #1: Yes

Reviewer #2: Yes

3. Have the authors made all data underlying the findings in their manuscript fully available?

Reviewer #1: Yes

Reviewer #2: Yes

4. Is the manuscript presented in an intelligible fashion and written in standard English?

Reviewer #1: Yes

Reviewer #2: No

5. Review Comments to the Author

Reviewer #1: In this study, authors collected 90 sex-matched neonates to evaluate the coherence of related clinical parameters and found that maternal preeclampsia is closely related with the offspring risk of CHD. Actually, this is not the first time to investigate those two factors’ relationship. Former scholars also got the same conclusion. For this study, I think there are some problems need to be resolved:

1. Table 2, in preeclampsia group, the total number of PDA is 4, however if you counted the sum of three category (isolated, with PFO and ASD), the number is just 2 (just in with ASD). What is the problem? Is there a mistake?

2. In the discussion, authors said,” Though prematurity is associated with a higher prevalence of CHD globally, our study did not find a significant association between prematurity and the odds of having CHD.” How can they get this opinion? Because in table 1 we can see that mean gestational age of women with preeclampsia or normal pregnancy is significantly different. Neonates whose mother with preeclampsia are premature and have a high risk of CHD. So, I think authors cannot give this conclusion.

3. As the design of this study, have all the mothers been excluded the genetic disease of heart, have they all detected the TORCHES. How about the history of delivery? Will it affect the health of newborns? Like the age of the first pregnancy and times of pregnancy and delivery.

Reviewer #2: Reviewer Notes (also attached as PDF)

PROFILE OF CONGENITAL HEART DISEASE IN INFANTS BORN FOLLOWING EXPOSURE TO PREECLAMPSIA

Summary:

This cohort study investigated the association between CHD in offspring of women with PE/E. They took into account for newborn sex, gestational age, and maternal factors (weight, age, and education). They reported that newborns from PE/E pregnancies were associated with an approximate 8 fold increase in risk for CHD. Most common of these defects included ASD, PFO, VSD, and PDA. Most of the PDA cases were resolved by the evaluation at day 28. These findings of this report is consistent with findings from other reports in observing increased CHD and decreased birth weight associated with PE/E pregnancies. However, this study re-evaluates the prevalence of PDA in infants from PE/E pregnancies. This study is unique in that it conducts a day 28 evaluation for CHD, therefore, accounting for PDA and PFO that may be transiently present in preterm infants. Overall, this paper provides statistical information about the association between PE/E pregnancies and increased CHD risk, and provides grounds for urging CHD screening for fetuses and newborns from PE/E pregnancies. Reviewers concerns are mentioned below with specific sections. The grammar throughout the paper needs to be reviewed with care.

Major Concerns:

In the abstract results section, you mention, “ASD (4 newborns), PDA (21 newborns), patent foramen ovale (14 newborns) and VSD (2 newborns) were the prevalent lesions in the first week of life”. The group (PE/E) to which this is referring to needs clarification.

In the abstract results section, you mention, “Being the infant of a woman with preeclampsia was associated with about 8-fold increased risk of having CHD (OR=7.9, 95%CI=2.5-24.9)”. If the p-value is mentioned it would provide more information about statistical significance.

In the introduction, you mentioned that one of the reasons to look into the association between CHD and PE/E is because PE/E itself entails an inflammatory state. However, before this statement is given, the etiology is not given. The reasoning could be more cohesive if a statement or two about PE/E and inflammation is provided in the first paragraph of this section.

In the results section, you mention “All of the PDA in newborns of normal pregnancy and 8 (8.9%) of those newborns of preeclamptic women had closed by the 28th day of life (Table 2)”. Table 2 shows that there are still 4 newborns (from PE/E) and 1 newborn (from normal pregnancy) with PDA. I believe it would be more accurate to say “most of the PDA cases” instead of “all”.

In the results section, you mention, “Being the infant of a woman with preeclampsia was associated with an ~8-fold increased likelihood of having CHD (OR=7.9, 95%CI=2.5-24.9). Being a female infant or being preterm was associated with an increased likelihood of having CHD of 1.8 times (OR=1.8, 95%CI=0.63-4.9) and 1.4 times (95% CI= 0.22-9.3) times respectively though not statistically significant”. Including actual p-values when describing the statistics would backup your statement.

In the discussion section, you mention, “This risk is dose dependent and likely due to a shared angiogenic imbalance in the mother and fetus.20-22”. What is the dose referring to? What is currently mentioned does not match the other statements or logic.

Minor Concerns:

Grammar errors throughout the paper. Please double check on the adverbs, verb tenses, subject verb agreement, conjunction usage, comma usage (often missing), and subject association.

Were the maternal nutritional status during pregnancy accounted for? Maternal nutrient status of various vitamins and minerals have a profound effect on fetal development.

It might be more informative to report the maternal weight in the form of BMI to account for their height.

PLOS One Questions:

What are the main claims of the paper and how significant are they for the discipline? Answer: Mentioned is summary (above).

Are the claims properly placed in the context of the previous literature? Have the authors treated the literature fairly? Answer: The connection between inflammation and PE/E in the introduction section needs to be elaborated. This study could prompt further investigation on the link between PE/E and CHD if the authors elaborate on potential mechanisms of angiogenic imbalance that leads to CHD development.

Do the data and analyses fully support the claims? If not, what other evidence is required? Answer: Yes, all of their claims are based on observed results and statistics.

PLOS ONE encourages authors to publish detailed protocols and algorithms as supporting information online. Do any particular methods used in the manuscript warrant such treatment? If a protocol is already provided, for example for a randomized controlled trial, are there any important deviations from it? If so, have the authors explained adequately why the deviations occurred? Answer: They utilized standardized diagnostic protocols for CHD. Statistical analysis methods are also provided.

If the paper is considered unsuitable for publication in its present form, does the study itself show sufficient potential that the authors should be encouraged to resubmit a revised version? Answer: This report needs major grammar editing before final submission.

Are original data deposited in appropriate repositories and accession/version numbers provided for genes, proteins, mutants, diseases, etc.? Answer: The author declares that all data collected from the study is made available to the public.

Are details of the methodology sufficient to allow the experiments to be reproduced? Answer: Yes, sufficient methods are provided.

Is the manuscript well organized and written clearly enough to be accessible to non-specialists? Answer: Yes, very clear organization. However, needs major grammar editing for smooth reading.

6. PLOS authors have the option to publish the peer review history of their article (what does this mean?). If published, this will include your full peer review and any attached files.

Reviewer #1: No

Reviewer #2: No

---

## [Author Response · Author response to Decision Letter 0]

29 Jan 2020

Reviewer number 1

1. Table 2, in preeclampsia group, the total number of PDA is 4, however if you counted the sum of three category (isolated, with PFO and ASD), the number is just 2 (just in with ASD). What is the problem? Is there a mistake?

Response:

The total number of PDA is 4 (2 with isolated lesions, 2 associated with ASD). It was a clerical error.

2. In the discussion, authors said,” Though prematurity is associated with a higher prevalence of CHD globally, our study did not find a significant association between prematurity and the odds of having CHD.” How can they get this opinion? Because in table 1 we can see that mean gestational age of women with preeclampsia or normal pregnancy is significantly different. Neonates whose mother with preeclampsia are premature and have a high risk of CHD. So, I think authors cannot give this conclusion.

Response:

Even though women with Preeclampsia have a significantly lower gestational age, our results in Table 3 showed no significant association between prematurity and risk of CHD. This is why we made the statement above.

3. As the design of this study, have all the mothers been excluded the genetic disease of heart, have they all detected the TORCHES. How about the history of delivery? Will it affect the health of newborns? Like the age of the first pregnancy and times of pregnancy and delivery.

Response:

None of the women had a clinical history of TORCHES. However, because of the peculiarity of our developing economy, pregnant women do not have genetic studies done routinely. In addition, the scope of our study did not include genetic study. Timing of delivery, mode of delivery and parity of the women was not significantly associated nor did it differ between the two cohorts. But we will in future look in that direction. 

 Reviewer 2

Major Concerns:

1. In the abstract results section, you mention, “ASD (4 newborns), PDA (21 newborns), patent foramen ovale (14 newborns) and VSD (2 newborns) were the prevalent lesions in the first week of life”. The group (PE/E) to which this is referring to needs clarification.

Response:

This statement makes reference to the overall spectrum of CHD in the total population and not in women with Preeclampsia. The statement about PE/E ends with the previous sentence.

2. In the abstract results section, you mention, “Being the infant of a woman with preeclampsia was associated with about 8-fold increased risk of having CHD (OR=7.9, 95%CI=2.5-24.9)”. If the p-value is mentioned it would provide more information about statistical significance.

Response:

Noted and corrected

3. In the introduction, you mentioned that one of the reasons to look into the association between CHD and PE/E is because PE/E itself entails an inflammatory state. However, before this statement is given, the etiology is not given. The reasoning could be more cohesive if a statement or two about PE/E and inflammation is provided in the first paragraph of this section.

Response:

Noted and corrected.

4. In the results section, you mention “All of the PDA in newborns of normal pregnancy and 8 (8.9%) of those newborns of preeclamptic women had closed by the 28th day of life (Table 2)”. Table 2 shows that there are still 4 newborns (from PE/E) and 1 newborn (from normal pregnancy) with PDA. I believe it would be more accurate to say “most of the PDA cases” instead of “all”.

Response:

Noted and corrected to read “The PDA in newborn….”

5. In the results section, you mention, “Being the infant of a woman with preeclampsia was associated with an ~8-fold increased likelihood of having CHD (OR=7.9, 95%CI=2.5-24.9). Being a female infant or being preterm was associated with an increased likelihood of having CHD of 1.8 times (OR=1.8, 95%CI=0.63-4.9) and 1.4 times (95% CI= 0.22-9.3) times respectively though not statistically significant”. Including actual p-values when describing the statistics would backup your statement.

Response:

Noted and corrected.

In the discussion section, you mention, “This risk is dose dependent and likely due to a shared angiogenic imbalance in the mother and fetus.20-22”. What is the dose referring to? What is currently mentioned does not match the other statements or logic.

Response:

The statement is hereby deleted.

Minor Concerns:

1. Grammar errors throughout the paper. Please double check on the adverbs, verb tenses, subject verb agreement, conjunction usage, comma usage (often missing), and subject association.

Response:

Noted with thanks. Will correct that.

2. Were the maternal nutritional status during pregnancy accounted for? Maternal nutrient status of various vitamins and minerals have a profound effect on fetal development.

It might be more informative to report the maternal weight in the form of BMI to account for their height.

Response:

Maternal nutritional status assay of vitamins and other nutrients was not done.

Once again, we appreciate the opportunity given to us to submit our manuscript. We also appreciate the great work done by the reviewers in reviewing our manuscript.

---

## [Decision Letter · Decision Letter 1]

14 Feb 2020

PONE-D-19-30623R1

PROFILE OF CONGENITAL HEART DISEASE IN INFANTS BORN FOLLOWING EXPOSURE TO PREECLAMPSIA

PLOS ONE

Dear Dr. Yilgwan,

Thank you for submitting your manuscript to PLOS ONE. After careful consideration, we feel that it has merit but does not fully meet PLOS ONE’s publication criteria as it currently stands. Therefore, we invite you to submit a revised version of the manuscript that addresses the points raised during the review process.

We would appreciate receiving your revised manuscript by 3/5/2020. To enhance the reproducibility of your results, we recommend that if applicable you deposit your laboratory protocols in protocols.io, where a protocol can be assigned its own identifier (DOI) such that it can be cited independently in the future. For instructions see: http://journals.plos.org/plosone/s/submission-guidelines#loc-laboratory-protocols

We look forward to receiving your revised manuscript.

Kind regards,

Linglin Xie

Academic Editor

PLOS ONE

Reviewers' comments:

Reviewer's Responses to Questions

**Comments to the Author**

1. If the authors have adequately addressed your comments raised in a previous round of review and you feel that this manuscript is now acceptable for publication, you may indicate that here to bypass the “Comments to the Author” section, enter your conflict of interest statement in the “Confidential to Editor” section, and submit your "Accept" recommendation.

Reviewer #1: (No Response)

Reviewer #2: (No Response)

2. Is the manuscript technically sound, and do the data support the conclusions?

Reviewer #1: Yes

Reviewer #2: Yes

3. Has the statistical analysis been performed appropriately and rigorously? 

Reviewer #1: Yes

Reviewer #2: Yes

4. Have the authors made all data underlying the findings in their manuscript fully available?

Reviewer #1: Yes

Reviewer #2: Yes

5. Is the manuscript presented in an intelligible fashion and written in standard English?

Reviewer #1: Yes

Reviewer #2: Yes

6. Review Comments to the Author

Reviewer #1: (No Response)

Reviewer #2: Reviewer 2 - this section also attached as word doc.

Major Concerns:

1. In the abstract results section, you mention, “ASD (4 newborns), PDA (21 newborns), patent foramen ovale (14 newborns) and VSD (2 newborns) were the prevalent lesions in the first week of life”. The group (PE/E) to which this is referring to needs clarification.

a. Response: This statement makes reference to the overall spectrum of CHD in the total population and not in women with Preeclampsia. The statement about PE/E ends with the previous sentence.

2nd Review: I understand now. But may be helpful to clarify by saying, “ASD (4 newborns), PDA (21 newborns), patent foramen ovale (14 newborns) and VSD (2 newborns) were the prevalent lesions among newborns of women without preeclampsia in the first week of life.”

2. In the abstract results section, you mention, “Being the infant of a woman with preeclampsia was associated with about 8-fold increased risk of having CHD (OR=7.9, 95%CI=2.5-24.9)”. If the p-value is mentioned it would provide more information about statistical significance.

a. Response: Noted and corrected

2nd Review: Verified.

3. In the introduction, you mentioned that one of the reasons to look into the association between CHD and PE/E is because PE/E itself entails an inflammatory state. However, before this statement is given, the etiology is not given. The reasoning could be more cohesive if a statement or two about PE/E and inflammation is provided in the first paragraph of this section.

a. Response: Noted and corrected.

2nd Review: Verified. I think the current version provides a very detailed background and justification for the study.

4. In the results section, you mention “All of the PDA in newborns of normal pregnancy and 8 (8.9%) of those newborns of preeclamptic women had closed by the 28th day of life (Table 2)”. Table 2 shows that there are still 4 newborns (from PE/E) and 1 newborn (from normal pregnancy) with PDA. I believe it would be more accurate to say “most of the PDA cases” instead of “all”.

a. Response: Noted and corrected to read “The PDA in newborn….”

2nd Review: Verified.

5. In the results section, you mention, “Being the infant of a woman with preeclampsia was associated with an ~8-fold increased likelihood of having CHD (OR=7.9, 95%CI=2.5-24.9). Being a female infant or being preterm was associated with an increased likelihood of having CHD of 1.8 times (OR=1.8, 95%CI=0.63-4.9) and 1.4 times (95% CI= 0.22-9.3) times respectively though not statistically significant”. Including actual p-values when describing the statistics would backup your statement.

a. Response: Noted and corrected.

2nd Review: Verified.

6. In the discussion section, you mention, “This risk is dose dependent and likely due to a shared angiogenic imbalance in the mother and fetus.20-22”. What is the dose referring to? What is currently mentioned does not match the other statements or logic.

a. Response: The statement is hereby deleted.

2nd Review: This statement is still there. What do you mean by dose? “This risk is dose dependent and likely due to a shared angiogenic imbalance in the mother and fetus.20-22”. The sentence after that also seems to not flow well right after this statement. The statement itself sounds a little redundant. Can you make the last few sentences of this paragraph more concise?

Minor Concerns:

1. Grammar errors throughout the paper. Please double check on the adverbs, verb tenses, subject verb agreement, conjunction usage, comma usage (often missing), and subject association.

a. Response: Noted with thanks. Will correct that.

2nd Review: Did not notice any major grammar issues.

2. Were the maternal nutritional status during pregnancy accounted for? Maternal nutrient status of various vitamins and minerals have a profound effect on fetal development. It might be more informative to report the maternal weight in the form of BMI to account for their height.

a. Response: Maternal nutritional status assay of vitamins and other nutrients was not done. Once again, we appreciate the opportunity given to us to submit our manuscript. We also appreciate the great work done by the reviewers in reviewing our manuscript.

2nd Review: Reporting the weight status of the mothers could be more informative if the weight was reported as BMI if available.

7. PLOS authors have the option to publish the peer review history of their article (what does this mean?). If published, this will include your full peer review and any attached files.

Reviewer #1: No

Reviewer #2: No

---

## [Author Response · Author response to Decision Letter 1]

17 Feb 2020

Reviewer 2 

Major Concerns: 

1. In the abstract results section, you mention, “ASD (4 newborns), PDA (21 newborns), patent foramen ovale (14 newborns) and VSD (2 newborns) were the prevalent lesions in the first week of life”. The group (PE/E) to which this is referring to needs clarification. 

2nd Review: I understand now. But may be helpful to clarify by saying, “ASD (4 newborns), PDA (21 newborns), patent foramen ovale (14 newborns) and VSD (2 newborns) were the prevalent lesions among newborns of women without preeclampsia in the first week of life.”

Second review response:

Corrected to read; Overall, ASD (4 newborns), PDA (21 newborns), patent foramen ovale (14 newborns) and VSD (2 newborns) were the prevalent lesions found among all the newborns studied in the first week of life.

2. In the discussion section, you mention, “This risk is dose dependent and likely due to a shared angiogenic imbalance in the mother and fetus.20-22”. What is the dose referring to? What is currently mentioned does not match the other statements or logic. 

a. Response: The statement is hereby deleted. 

2nd Review: This statement is still there. What do you mean by dose? “This risk is dose dependent and likely due to a shared angiogenic imbalance in the mother and fetus.20-22”. The sentence after that also seems to not flow well right after this statement. The statement itself sounds a little redundant. Can you make the last few sentences of this paragraph more concise? 

Second review answer:

Sentence rephrased.

Minor Concerns: 

1. Were the maternal nutritional status during pregnancy accounted for? Maternal nutrient status of various vitamins and minerals have a profound effect on fetal development. It might be more informative to report the maternal weight in the form of BMI to account for their height. 

a. Response: Maternal nutritional status assay of vitamins and other nutrients was not done. Once again, we appreciate the opportunity given to us to submit our manuscript. We also appreciate the great work done by the reviewers in reviewing our manuscript.

2nd Review: Reporting the weight status of the mothers could be more informative if the weight was reported as BMI if available. 

Response:

Done

---

## [Editor Report · Decision Letter 2]

20 Feb 2020

PROFILE OF CONGENITAL HEART DISEASE IN INFANTS BORN FOLLOWING EXPOSURE TO PREECLAMPSIA

PONE-D-19-30623R2

Dear Dr. Yilgwan,

We are pleased to inform you that your manuscript has been judged scientifically suitable for publication and will be formally accepted for publication once it complies with all outstanding technical requirements.

With kind regards,

Linglin Xie

Academic Editor

PLOS ONE
---

## [Editor Report · Acceptance letter]

2 Mar 2020

PONE-D-19-30623R2 

PROFILE OF CONGENITAL HEART DISEASE IN INFANTS BORN FOLLOWING EXPOSURE TO PREECLAMPSIA 

Dear Dr. Yilgwan:

I am pleased to inform you that your manuscript has been deemed suitable for publication in PLOS ONE. Congratulations! Your manuscript is now with our production department. 

With kind regards,

on behalf of

Dr. Linglin Xie 

Academic Editor

PLOS ONE